# Dosimetric Impact of Lesion Number, Size, and Volume on Mean Brain Dose with Stereotactic Radiosurgery for Multiple Brain Metastases

**DOI:** 10.3390/cancers15030780

**Published:** 2023-01-27

**Authors:** Alonso La Rosa, D Jay J. Wieczorek, Ranjini Tolakanahalli, Yongsook C. Lee, Tugce Kutuk, Martin C. Tom, Matthew D. Hall, Michael W. McDermott, Minesh P. Mehta, Alonso N. Gutierrez, Rupesh Kotecha

**Affiliations:** 1Department of Radiation Oncology, Miami Cancer Institute, Baptist Health South Florida, Miami, FL 33176, USA; 2Department of Radiation Oncology, Herbert Wertheim College of Medicine, Florida International University, Miami, FL 33199, USA; 3Department of Neurosurgery, Miami Cancer Institute, Baptist Health South Florida, Miami, FL 33176, USA; 4Department of Translational Medicine, Herbert Wertheim College of Medicine, Florida International University, Miami, FL 33199, USA

**Keywords:** radiosurgery, GammaKnife, inverse optimizer, CyberKnife, multiple brain metastases, mean brain dose

## Abstract

**Simple Summary:**

There is an increasing incidence of patients diagnosed with multiple brain metastasis (MBM) in the modern era. Although stereotactic radiosurgery (SRS) has been shown to result in similar survival with less neurocognitive deterioration for patients with MBM, the intracranial disease extent, both in quantity and volume, able to be treated with modern dedicated SRS platforms has yet to be empirically demonstrated. In this study, we evaluate the effect of the number and volume of lesions treated on mean brain dose using two dedicated intracranial SRS delivery technologies. We found that mean brain dose linearly increased with the number of lesions and total gross tumor volume (GTV), while selected metrics associated with radiation necrosis risk (i.e., V8 Gy, V10 Gy, and V12 Gy) had quadratic correlations to the number of lesions and the total GTV. This study shows that it is dosimetrically feasible to treat MBMs with SRS.

**Abstract:**

We evaluated the effect of lesion number and volume for brain metastasis treated with SRS using GammaKnife^®^ ICON™ (GK) and CyberKnife^®^ M6™ (CK). Four sets of lesion sizes (<5 mm, 5–10 mm, >10–15 mm, and >15 mm) were contoured and prescribed a dose of 20 Gy/1 fraction. The number of lesions was increased until a threshold mean brain dose of 8 Gy was reached; then individually optimized to achieve maximum conformity. Across GK plans, mean brain dose was linearly proportional to the number of lesions and total GTV for all sizes. The numbers of lesions needed to reach this threshold for GK were 177, 57, 29, and 10 for each size group, respectively; corresponding total GTVs were 3.62 cc, 20.37 cc, 30.25 cc, and 57.96 cc, respectively. For CK, the threshold numbers of lesions were 135, 35, 18, and 8, with corresponding total GTVs of 2.32 cc, 12.09 cc, 18.24 cc, and 41.52 cc respectively. Mean brain dose increased linearly with number of lesions and total GTV while V8 Gy, V10 Gy, and V12 Gy showed quadratic correlations to the number of lesions and total GTV. Modern dedicated intracranial SRS systems allow for treatment of numerous brain metastases especially for ≤10 mm; clinical evidence to support this practice is critical to expansion in the clinic.

## 1. Introduction

Brain metastases (BM) are the most common intracranial tumors diagnosed in adults, occurring in one of every three oncologic patients. Increasingly, with the transition from CT to MRI-based screening and enhanced diagnostic imaging using dedicated thin-slice MRI sequences, patients are more commonly diagnosed with multiple brain metastases (MBM) in clinical practice [1,2].

The management of patients with MBM has evolved over time. Although historically, these patients were treated with whole brain radiation therapy (WBRT), more recently, patients with MBM and high-performance status at presentation may be treated with hippocampal avoidant WBRT (HA-WBRT) to reduce the risk of neurocognitive decline (NCD) following treatment. Using a HA-WBRT approach, the risk of NCD was less in comparison to WBRT (HR 0.74; 95%; CI 0.58–0.95, *p* = 0.02), however was still observed in >50% of patients [3,4]. Furthermore, the neurocognitive difference emerged starting at the fourth month, a concern for the vast majority of brain metastasis survivors over 6 months. Therefore, the significant proportion as well as early incidence of neurocognitive deterioration has led to a paradigm shift of treating patients with MBM to be considered for SRS [4].

Primary SRS for patients with one to four brain metastases has been accepted into national guidelines based on randomized data [5,6,7,8]. Additionally, based on prospective, nonrandomized data, SRS alone for five to ten tumors and/or a total volume of up to 15 mL may be associated with favorable outcomes in terms of local control and toxicity [6,7,9]. More recently, randomized evidence exists to support SRS alone for patients with up to 15 intracranial lesions, given the similar survival and reduced risk of neurocognitive dysfunction compared to WBRT [10]. A phase III randomized trial studying SRS vs HA-WBRT for 5 to 20 metastases (NCT03075072) is still under accrual and will provide further prospective evidence to guide clinical practice. Despite this clinical trial eligibility, numerous retrospective series have described the outcomes of treating MBM beyond these thresholds with favorable local control rates and modest treatment-related toxicities [11,12,13,14]. Despite these reports in clinical practice, the current SRS limits with regard to lesion number and volume with modern dedicated SRS platforms have yet to be systemically studied and empirically demonstrated.

In this study we aim to evaluate the effect of the number and volume of lesions based on individual metastases size planned with SRS on mean dose to the brain. We used two dedicated photon radiosurgery systems—Gamma Knife^®^ (Elekta, Stockholm, Sweden) and CyberKnife^®^ (Accuray, Madison, WI, USA). The Gamma Knife^®^ (GK) Icon™ system used in this study consists of eight movable sectors with a total of 192 _60_Co sources, where the sectors can be set to four different collimator settings (4, 8, and 16 mm in diameter) [15]. CyberKnife^®^ M6™ (CK) utilizes a LINAC mounted on a robotic arm to deliver radiation from hundreds of potential angles [16]. Both of these systems have a longstanding history and published clinical evidence with SRS for MBM. Using dosimetric surrogates for radiation necrosis risk—mean brain dose and the volume of the brain receiving at least 8 Gy, 10 Gy, or 12 Gy—we evaluated the theoretical potential for treatment of MBM with single fraction SRS on these two SRS delivery systems.

## 2. Materials and Methods

Four sizes of GTVs were contoured and grouped by lesion size: Group 1 (<5 mm), Group 2 (5–10 mm), Group 3 (>10–15 mm), and Group 4 (>15 mm). Group sizes were selected in finer 5 mm increments to provide better dosimetric discrimination and help highlight any differences based on delivery technology, similar to previous series demonstrating differences in clinical outcomes with these increments. [17] GTVs were contoured mimicking clinical scenarios with some targets spaced far apart and some clustered together within 1 mm distance. The distance to the closest neighboring target in three dimensions (3D) was then measured and tabulated for each target. Volumes contoured varied from spherical to ellipsoidal in shape resembling clinical targets. Targets were then uniformly distributed throughout the whole brain. A normal brain structure inclusive of the GTVs was also contoured. The reference brain CT image set was selected as it represented an average head CT data set. The physical dimensions of the head, volume of brain, visualization of anatomical structures and spatial resolution (i.e., < 1 mm voxels) of CT images were well suited for SRS planning. Additionally, the CT image data set was free of metal artifacts. SRS plans were generated in the treatment planning systems (TPS) for both GK (Leksell GammaPlan^®^ Version 11.3.2) and CK (Accuray Precision™ Version 3.3.1.2), respectively, and prescribed to >99.5% of the GTV receiving 20 Gy in one fraction. For GK, treatment plans were generated using the Fast Inverse Planning (FIP) dose optimizer [18], commercially referred to as Lightning™, with the 0.8 low dose (LD) and 0.3 beam on time (BOT) optimization settings for Group 1, 0.7 LD/0.4 BOT settings for Groups 2 and 3, and 0.65 LD/0.4 BOT setting for Group 4 and with the coverage option enabled. These settings were selected based on prior work testing the Lightning Optimizer and used in this study to reduce inter-physicist variability in treatment planning [18]. Dose prescription isodose lines were limited to at least 50% for both GK and CK plans. Resulting plans were then manually optimized to achieve a maximum conformity for each target. For CK planning, SRS plans were generated using cones calculated with the VOLO optimizer and the Monte Carlo dose engine. The dose calculation settings were set to high resolution and 1% uncertainty. Dose was then prescribed such that there was at least 99.6% coverage for all targets.

GK and CK plans for each size group were initially developed with a nominal number of lesions, randomly selected to be widely distributed in the brain. The number of lesions per plan was then progressively increased until a mean threshold dose of approximately 8 Gy to the whole brain (including the GTVs) was reached for the composite plan. Of note, a mean single fraction equivalent dose (SFED) in WBRT, according to the linear quadratic (LQ) model, is approximately 10 Gy. However, this prescribed single dose was associated with 6.7% of deaths shortly following treatment, is above the tolerance of optic structures [19,20], and was also abandoned in prior ultra-rapid, high-dose WBRT clinical trials, thus a reduced threshold of 8 Gy was used in this planning study as the threshold mean brain dose [21]. The brain volumes receiving at least 8, 10, and 12 Gy (V8 Gy, V10 Gy, and V12 Gy), mean brain dose, total GTV volumes, and the total number of lesions were also tabulated for each plan generated. As the maximum number of targets in any given plan is limited to 52 in the GK planning system, composite doses were tabulated in a third-party software, Velocity™ (Varian Medical Systems, Palo Alto, CA, USA). In addition, for GK planning, a second set of data was also generated using targets in order of increasing volume for Groups 3–4 and for a second set of random ordering of targets for Groups 1–2. GK plans for Groups 2 and 3 were also analyzed before (Raw) and after manual adjustment/optimization (Optimized) for maximum target conformity. Finally, we compared V8 Gy, V10 Gy, and V12 Gy for Groups 2 and 3 by the treatment planning system used (GK versus CK).

The above metrics were plotted as a function of number of lesions and total volume of lesions, and correlation coefficient (R^2^) was calculated.

## 3. Results

### 3.1. Target Distribution

The distance of the closest neighbor for each target was measured in 3D and tabulated in Table 1. The distribution of targets for Groups 1–4, shown in Figure 1a–d, shows some targets are widely spaced and some located in clusters as found in clinical settings. The GTVs are illustrated in 3D in Figure 2a–h.

### 3.2. Effect of the Number of Lesions and Total Volume

#### 3.2.1. Brain Mean Dose

The range of volumes and number of lesions that resulted in the mean brain dose of approximately 8 Gy in a single fraction for each size group, including target volumes and maximum target dimensions, are shown in Table 2 for GK and CK.

The number of lesions needed to generate a mean dose of 8 Gy in one fraction equivalence in GK was 177 in Group 1, 57 in Group 2, 29 in Group 3, and approximately 10 in Group 4. This corresponded to total GTVs of 3.62 cc, 20.37 cc, 30.25 cc, and 57.96 cc, respectively. For CK, the number of lesions needed to generate a mean dose of 8 Gy in one fraction equivalence was 135 for Group 1, 35 for Group 2, 18 for Group 3, and 8 for Group 4, corresponding to total GTVs of 2.32 cc, 12.09 cc, 18.24 cc, and 41.52 cc, respectively.

The disposition of mean brain dose according to the number of lesions and total GTVs for these four groups, is shown for GK in Figure 3 and for CK in Figure 4. As observed, the mean brain dose increases linearly (correlation coefficient as in Figure 3 and Figure 4) as a function of the number of lesions and the total GTVs treated rises.

#### 3.2.2. Other Dosimetric Parameters (V8 Gy, V10 Gy and V12 Gy)

In addition, V8 Gy, V10 Gy, and V12 Gy values as a function of the number of lesions and the total GTVs for Groups 1–4 are shown in Figure 5a–f for GK and Figure 6a–f for CK.

The V8 Gy, V10 Gy, and V12 Gy curves for both GK and CK were found to have a quadratic correlation with the number of lesions treated and the total GTVs (R^2^ as shown in Figure 5 and Figure 6). The V8 Gy, V10 Gy, and V12 Gy tend to exhibit a steeper increase for larger lesions (Groups 3 and 4) as the number of lesions treated increases. On the other hand, for the smallest lesions size (Group 1), similar steeper increase is noted while total GTVs increases. This trend is true for both GK and CK as shown in Figure 5 and Figure 6, respectively.

### 3.3. Effect of Treatment Planning Technique

The mean dose, V8 Gy, V10 Gy, and V12 Gy values are found to be highly dependent on planning technique; these results are shown in Figure 7, Figure 8 and Figure 9, where we can appreciate that the V8 Gy, V10 Gy, and V12 Gy values are affected by manual optimization of treatment plans for maximum conformity, order of target treatment (random or increasing volume), and by treatment platform.

## 4. Discussion

MBM are increasingly diagnosed in clinical practice given improvements in diagnostic imaging [2]. Although WBRT—either conventionally delivered or with use of hippocampal avoidance technique—is utilized commonly, the additional acute toxicities, overall length of treatment time, need to hold systemic therapies during radiotherapy treatment, and neurocognitive risk, have led to a shift towards primary SRS as an alternative [3]. Although treatment of numerous brain metastasis beyond standard thresholds (10–15 lesions) have been published, the true maximum number of lesions or total volume that can be treated with modern SRS platforms while respecting dosimetric constraints has not been systematically evaluated. Therefore, this study evaluated the effect of lesion size and number on mean brain dose and several metrics associated with radiation necrosis risk (V8 Gy, V10 Gy, and V12 Gy) for a model case planned for SRS, prescribed to a dose of 20 Gy in one fraction. Several important conclusions can be drawn from this work. First, the number of lesions treated, grouped by size to reach a mean brain dose of 8 Gy, for GK was between 10 to 177 lesions, and for CK between 8 to 135 lesions. We also characterized how this can change by varying the lesion volume and determined that for GK, from 3.32 to 57.955 cc could be treated and for CK, from 2.322 to 41.524 cc could be treated. Second, in this study we were able to characterize how lesion number and volume affected those metrics. For mean brain dose, a linear relationship was established, however, a quadratic relation with the V8 Gy, V10 Gy, and V12 Gy parameters was observed with lesion number and volume.

Based on the current NCCN guidelines (version 5.2022), patients with limited or multiple brain metastases can be treated with SRS or WBRT alone. The ASTRO guidelines strongly suggest SRS alone in patients with a good performance status (ECOG 0-2), for up to four lesions, and consider it in patients with up to ten BM [22]. The ASCO/SNO guidelines suggest SRS alone for one to four unresected BM, excluding small cell carcinoma, and consider it as an option for more than four unresected BM in patients with good performance statuses (KPS ≥ 70) [9]. A majority of the evidence supporting these recommendations is derived from multiple trials in patients with one to four brain metastases; however, a recently presented trial [10] supported the use of primary SRS in patients with 4–15 untreated brain metastases, with up to 20 lesions at the time of the treatment allowed. In this trial, patients were stratified by number of lesions (4–7 vs. 8–15), and although the trial overall showed better cognitive composite scores in those who received SRS, subgroup analyses and long-term toxicity results are pending at this time.

In addition to the Level I evidence supporting the efficacy of SRS for those with limited disease, the JLGK0901 study provided Level II evidence supporting treatment of five to ten lesions using GK as long as the maximum diameter of any lesion did not exceed 3 cm, or the total intracranial volume did not exceed 15 cc. However, additional retrospective series have published outcomes in patients treated with 15 to 30 lesions with SRS alone [11,12,14,23]. Clearly, lesion size or volume is important to note when treating multiple brain metastases with SRS alone when considering dosimetric constraints. For example, punctate sized lesions (≤5 mm) require the largest number of lesions to generate the same mean brain dose compared to larger sized lesions as shown in Figure 3 and Figure 4, irrespective of the treatment modality. However, the largest number of lesions do not translate to the largest total GTV volume. Prior studies reporting which patients would benefit from SRS rather than WBRT for MBM, have suggested a volume cutoff of 10–15 cc [6,24,25,26,27], others had observed similar outcomes with total intracranial target volumes up to 30 cc [28,29]. Using the data generated from this study, a 10–15 cc cut off appears to be very conservative with an estimated mean brain dose of approximately 2 Gy. Even a 30 cc threshold with lesions measuring 5 mm or less each would only result in a mean brain dose of approximately 4 Gy. On the other hand, Group 4 lesions (>15 mm) in this study, showed the least number of lesions to generate an 8 Gy mean brain dose corresponding to the largest total GTV volume. Serizawa et al. reported the worst prognosis in terms of neurological deterioration and death in patients treated with GK SRS alone with a total volume ≥15 cc, large tumor size (≥2.5 cm), localized leptomeningeal disease and clinically neurological symptoms [30]. When classified by tumor diameter, grade 3+ neurotoxicity was 10% for ≤2 cm lesions treated to a peripheral dose of 24 Gy, 20% for 2.1–3 cm treated to 18 Gy, and 14% for lesions 3.1–4 cm treated to 15 Gy. In multivariate analyses, tumor diameter was associated with a significantly increased risk of grade 3+ neurotoxicity, with 7.3 and 16 fold increased risks for tumors 2.1–3 and 3.1–4 cm versus ≤2 cm [31]. Sita et al. recently published data from 30 patients treated with single fraction SRS for ten or more metastasis from different solid tumor primary sites evaluating clinical and dosimetric outcomes based on GK treatment (median of 13 lesions ranging from 10 to 26). In their study, the mean dose to the brain was not related to the number of lesions (Pearson r = 0.23, *p* = 0.21), but was closely associated with total tumor volume (Pearson r = 0.95, *p* < 0.0001) [32]. Given that this is well below the threshold for significant mean brain dose, this is likely the reason no correlation was established in their study. For example, based on the results from our study, even with 5–10 mm sized lesions, one would be able to treat 57 with GK and 35 with CK to reach the mean brain dose threshold.

In addition to mean brain dose, we also evaluated several dosimetric parameters associated with radiation necrosis (RN). This complication can develop from a range of months to even several years after SRS. Select parameters from previous studies, historically used for establishing constraints to reduce the risk of RN are presented in Table 3 [13,31,33,34,35,36,37,38,39,40,41,42,43,44,45,46,47,48,49,50]. As seen, substantial variations exist between studies, with some using the whole brain volume, others using the diameter of the treated lesion, and yet others basing the risk on the treated volume. Therefore, consensus efforts to define tolerance doses with different dose fractionation schemes are clearly needed. One specific parameter, individual target V12 Gy, has been recently suggested as a unique dosimetric constraint as opposed to cumulative V12 Gy [51]. The V8 Gy, V10 Gy, and V12 Gy changes are characterized in Figure 5 and Figure 6 for GK and CK plans, respectively, and show a trend toward quadratic behavior. For the same brain volume, Group 1 shows the largest number of lesions corresponding to the least total GTV volume, while Group 4 shows the least number of targets treated with the largest total GTV volume.

There are many differences between GK, CK, and LINAC-based SRS delivery, but there are no clear indications for choosing one modality over the other; no clinical trials directly comparing GK and LINAC-based SRS have been published to date. In RTOG 9508, patients included in the experimental arm received an SRS boost with either GK or LINAC, however, no benefit of either system was established [5]. In terms of risk for RN, as seen in Table 3, outcomes are very heterogeneous between different techniques in different studies, preventing direct comparative analyses. In addition to differences between the platforms themselves, in this study, V8 Gy, V10 Gy, and V12 Gy volumes were found to also be affected by treatment planning techniques as shown in Figure 8, Figure 9 and Figure 10. For example, in Figure 8, Groups 2 and 3 show a marked increase in the V8 Gy, V10 Gy, and V12 Gy values when GK FIP plans were not optimized to maximize target conformity. Optimizing involves either choosing the maximum prescription isodose line to give >99.5% target coverage, moving the shot location, or changing shot sector collimator size to maximize target Paddick conformity index (PCI). Differences of 0.1–32.8% can be seen between a raw and an optimized GK plan and this tended to increase with the number of targets for both Groups 2 and 3. Ensuring a plan characterized by high PCIs and minimizing gradient index (GI) leads to lower brain doses. This is highly relevant in GK planning where each target can be optimized individually and can result in a highly conformal composite plan. Figure 8 and Figure 9 show how the order of GTV treatment selection can also affect the brain dose metric curve. Random ordering of the treated GTVs generates less variability in the dose metric curve as compared to an increasing volume ordering. These variabilities in the mean brain and V8 Gy to V12 Gy values were also found across the technologies. To this end, Figure 10 shows the dose metric curves for GK and CK for Groups 3 and 4. For the same total GTV treated, GK tends to treat less normal brain. Wowra et al., reported that besides the better intratumor homogeneity, CK was better in radiation protection in terms of a lower peripheral dose in comparison to GK, but only when treating a single lesion [52]. Therefore, future studies comparing risk of radiation necrosis between technologies should also account for differences in planning techniques as well.

In this study, we also demonstrated a significant difference in the number of maximum lesions and corresponding total volume treated between GK plans and CK plans. The large difference can be attributed to the difference in planning normalization between GK and CK. Whereas multi-target GK plan is a composite of all the individual target plans, each with their own prescription and maximized PCI, in CK planning, the plan is normalized to the least covered GTV in the entire plan to achieve at least 99.6% coverage of all GTV targets. This results in over-coverage of some targets to achieve the minimum dose to all targets. In this way, it is very difficult to maximize conformity of each target. This target over-coverage progressively gets worse as the number of targets increases and accounts for the significant difference between the V8 Gy, V10 Gy, and V12 Gy, the number of lesions and total GTV values of CK and GK. In addition, even between the same dose planning platform, dose metrics between plans generated with different techniques, i.e., different ordering in the treatment of the targets whether in random order or increasing volume size, exhibit slight variability as shown in Figure 8 and Figure 9. This finding can be used for comparison between with other systems such as other LINAC-based techniques, proton or newer SRS technologies for future SRS comparative benchmarking exercises.

The limitation of the study is that being non-patient based, no clinical outcomes were obtained, and is limited to a theoretical treatment planning exercise. We created plans based on a uniform distribution of lesions throughout the brain, however, potential clustering of lesions would result in differences to the observed metrics in this study. In terms of dosimetry, simultaneous optimization was not possible for GK for all lesions once >52 targets were planned. Due to the target definition limit of 52 in LGP, plans with targets >52 were split, and dose composite was created outside of LGP. This removes the final plan optimization capability for all targets in the plan. Additionally, any change on the distribution of lesions or mixed sized scenario may affect the relation (linear/quadratic) found, and this was not explored in this study.

## 5. Conclusions

We have found a linear relationship between the number of lesions and the total volume of all lesions treated (GTVs) with mean brain dose. This study also showed a quadratic relation between number of lesions and total GTVs with other parameters, such as V8 Gy, V10 Gy, and V12 Gy, depending on the planning technique used for maximizing conformity. The data showed a dosimetric feasibility for treating numerous lesions with the current technology, much beyond practice supported by current clinical trials but often performed in routine SRS practice. An important difference between the system used (GK versus CK) in terms of the amount (number and volume) of intracranial disease for SRS was also demonstrated in this study. Furthermore, the methods and data of this study provide a framework and dosimetric benchmark whereby additional dedicated SRS platforms for multiple brain metastases can be compared.

## Figures and Tables

**Figure 1 cancers-15-00780-f001:**
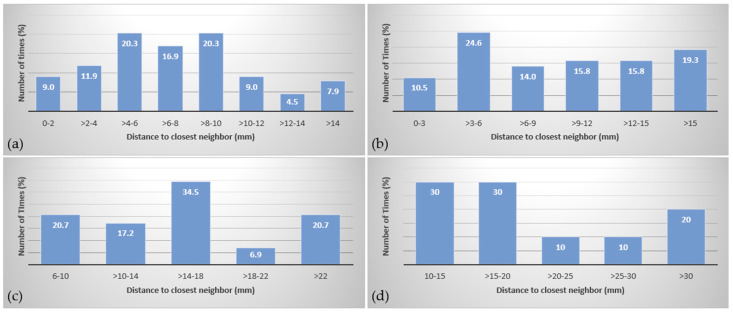
Distance to closest neighbor target distribution for Groups 1 (**a**), 2 (**b**), 3 (**c**) and 4 (**d**).

**Figure 2 cancers-15-00780-f002:**
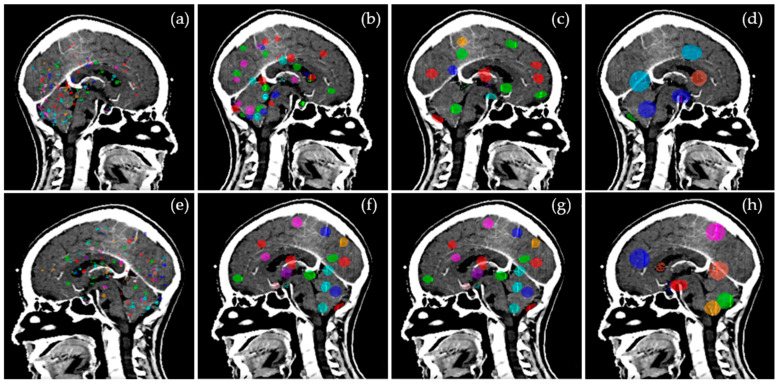
Target distribution for Groups 1–4 as viewed from the left (**a**–**d**) and right (**e**–**h**) of the patient.

**Figure 3 cancers-15-00780-f003:**
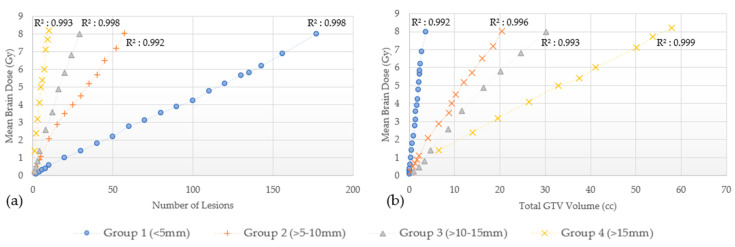
GammaKnife. Mean brain dose by lesion number (**a**) and by total GTVs (**b**). Linear correlation coefficient (R^2^).

**Figure 4 cancers-15-00780-f004:**
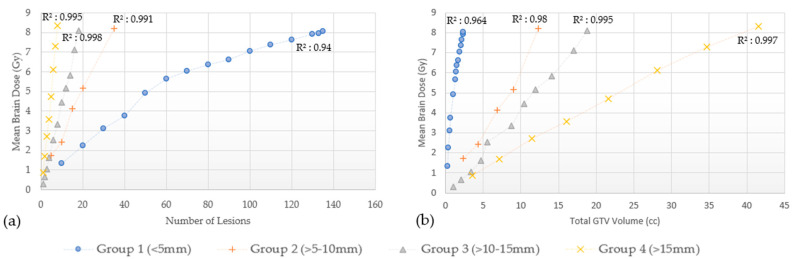
CyberKnife. Mean brain dose by lesion number (**a**) and by total GTVs (**b**). Linear correlation coefficient (R^2^).

**Figure 5 cancers-15-00780-f005:**
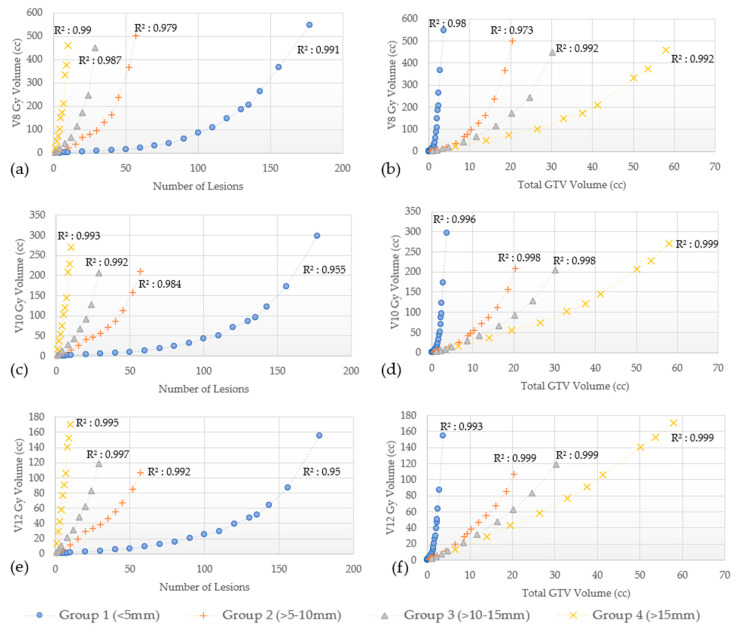
(**a**–**f**) V8 Gy, V10 Gy and V12 Gy vs number of lesions and total GTVs by lesion size group planned with GammaKnife. Quadratic correlation coefficient (R^2^).

**Figure 6 cancers-15-00780-f006:**
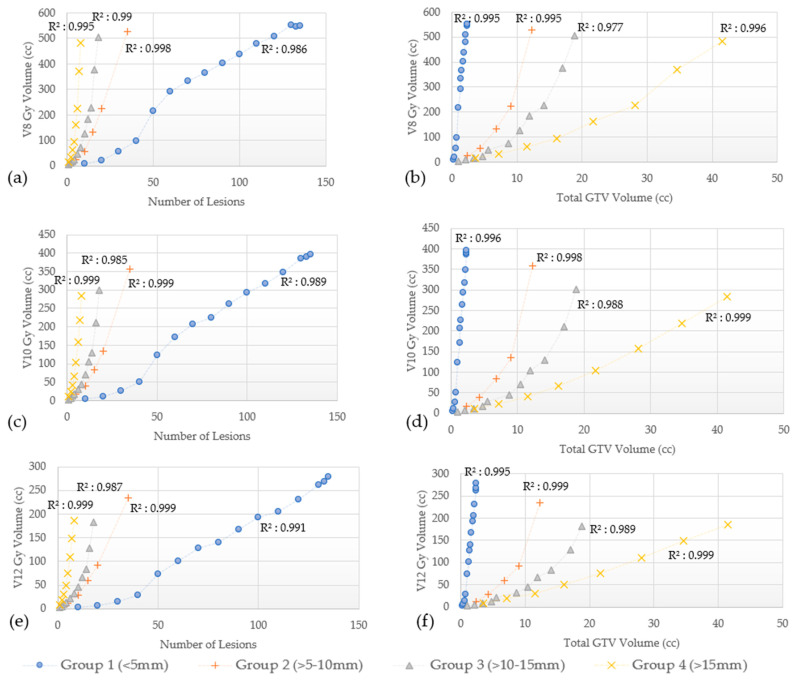
(**a**–**f**). V8 Gy, V10 Gy, and V12 Gy vs number of lesions and total GTVs by lesion size group planned with CyberKnife. Quadratic correlation coefficient (R^2^).

**Figure 7 cancers-15-00780-f007:**
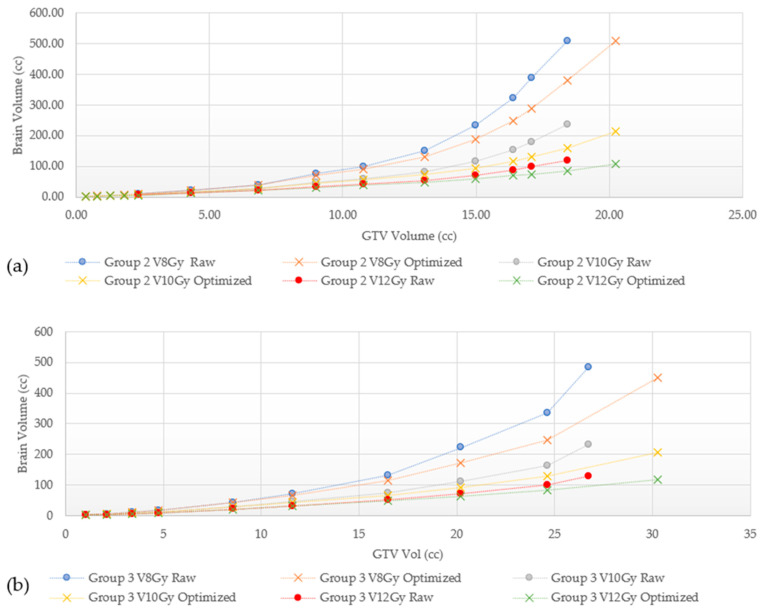
Comparing raw versus optimized plans in GammaKnife (**a**) V8 Gy, V10 Gy, and V12 Gy for Group 2 lesions (**b**) V8 Gy, V10 Gy, and V12 Gy for Group 3 lesions.

**Figure 8 cancers-15-00780-f008:**
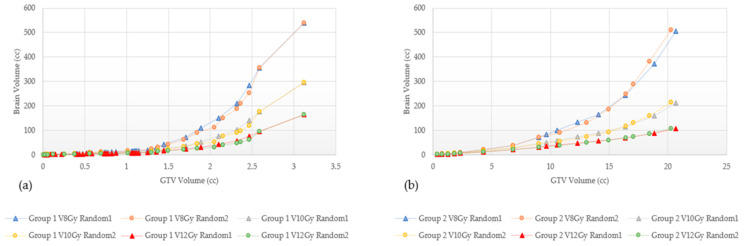
Comparison of V8 Gy, V10 Gy, and V12 Gy of two randomly ordered targets from Group 1 (**a**) and Group 2 (**b**) with GammaKnife.

**Figure 9 cancers-15-00780-f009:**
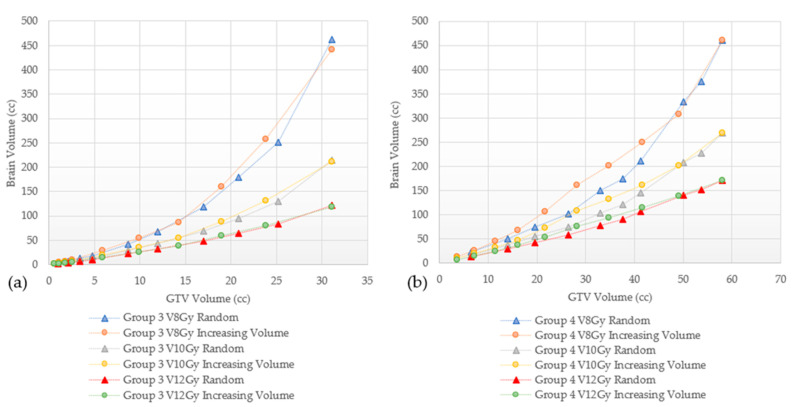
Comparison of V8 Gy, V10 Gy, and V12 Gy of randomly ordered targets and increasing volumes from Group 3 (**a**) and Group 4 (**b**) with GammaKnife.

**Figure 10 cancers-15-00780-f010:**
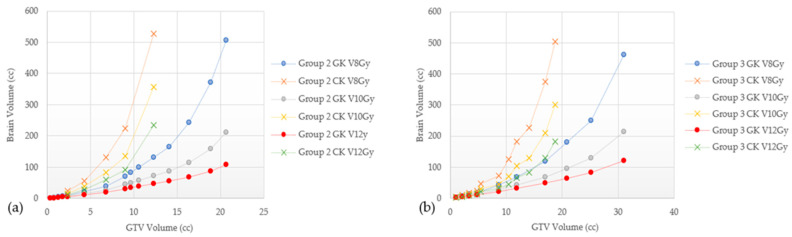
Comparison of V8 Gy, V10 Gy, and V12 Gy for CyberKnife and GammaKnife planning systems for Groups 2 (**a**) and 3 (**b**).

**Table 1 cancers-15-00780-t001:** Distance to the closest neighbor for each target in each group.

	Group 1 (<5 mm)	Group 2 (5–10 mm)	Group 3 (>10–15 mm)	Group 4 (>15 mm)
Mean Distance to Closest Neighbor (mm)	7.5	10.2	15.9	22.2
Median Distance to Closest Neighbor (mm)	7.1	9.5	15.1	19.6
Range Distance to Closest Neighbor (mm)	(0.5–22.7)	(1–32.8)	(6–28.3)	(13.5–43.9)

**Table 2 cancers-15-00780-t002:** Number of lesions and total volume grouped by size to reach a total brain mean dose of 8 Gy.

	Group 1 (<5 mm)		Group 2 (5–10 mm)		Group 3 (>10–15 mm)		Group 4 (>15 mm)	
	GK	CK	GK	CK	GK	CK	GK	CK
Number of lesions	177	135	57	35	29	18	10	8
Total GTV Volume (cc)	3.62	2.322	20.366	12.088	30.252	18.244	57.955	41.524
Mean GTV Volume (cc)	0.02 ± 0.017	0.017 ± 0.016	0.357 ± 0.141	0.352 ± 0.16	1.043 ± 0.259	1.014 ± 0.261	5.974 ± 1.776	5.188 ± 1.348
GTV Volume Range (cc)	(0.004, 0.081)	(0.001, 0.064)	(0.116, 0.619)	(0.116, 0.619)	(0.53, 1.567)	(0.53, 1.763)	(3.535, 8.922)	(3.535, 6.881)
Mean GTV Dimension (mm)	3.6 ± 1.0	3.4 ± 1.0	9.0 ± 1.1	8.9 ± 1.3	13.8 ± 1.1	13.8 ± 1.2	23.4 ± 2.1	23.0 ± 2.2
GTV Dimension Range (mm)	(1.8, 5)	(1.8, 5)	(6.1, 9.9)	(6.1, 9.9)	(10.8, 14.9)	(10.8, 14.9)	(19.8, 26.1)	(19.8, 26.1)
Median GTV Dimension (mm)	3.8	3.2	9.8	9.8	13.8	14.3	24.0	23.4

GK = Gamma Knife, CK = Cyber Knife, GTV = Gross tumor volume.

**Table 3 cancers-15-00780-t003:** Relationship between radiation necrosis risk and dose received by normal brain tissue, tumor diameter, and normal brain volume as published in prior SRS studies.

	Study	System Used	Parameter	Volume Constraint	RN Risk/Comments
Brain–normal tissue dose received	Miyawaki et al., 1999 (AVMs) [33]	LINAC non-CK	V16 Gy	>14 cc	72% MR changes; 22% RN resected
	Voges et al., 1996 (Mixed) [48]	LINAC non-CK	V10 Gy	>10 cc	23.7%
	Flickinger et al., 1997 (AVMs) [49]	GK	V12 Gy	-	10.7%
	Chin et al., 2001 (Mixed) [45]	GK	V10 Gy	-	-
	Koryto et al., 2006 (BM) [36]	GK	V12 Gy	>10 cc	>50%
	Blonigen et al., 2010 (BM) [38]	LINAC non-CK	V10 Gy	>10.5 cc	35%
			V12 Gy	7.85 cc	
			V8 Gy and V16 Gy	-	Showed most predictive for SRN (*p* < 0.0001)
	Minniti et al., 2011 (BM) [37]	LINAC non-CK	V10 Gy	>12.3 cc	47%
			V12 Gy	>10.9 cc	47%
			V12 Gy	6–10.9 cc	24%
	Ohtakara et al. 2012 (BM) [34]	LINAC non-CK	V15 Gy	5.20 cc	Presented as cut-off in patients with no prior WBRT
			V22 Gy	2.14 cc	
	Inoue et al., 2013 (BM) [47]	CK	V14 Gy	≥7 cc	12.8%—SFED (from 5 fractions)
	Inoue et al., 2014 (BM) [46]	CK	V14 Gy	≥7 cc	6.2%—SFED (from 3 fractions)
	Peng et al., 2019 (BM) [35]	-	V14 Gy	5 cc	0.4%
				10 cc	0.8%
				20 cc	3.4%
	Milano et al., 2021 (AVMs and BM) [31]	GK, LINAC	V12 Gy	5 cc	10%
			V12 Gy	10 cc	15%
			V12 Gy	>15 cc	20%
Diameter of BMs	Shaw et al., 2000 (BM, PBT) [39]	GK, LINAC	>2.1-4 cm		x7.3–16
	Kohutek et al., 2015 (BM) [40]	LINAC non-CK	>1.5 cm		37.5%
	Minniti et al., 2016 (BM) [51]	LINAC non-CK	>2 cm		20%
			>3 cm		33%
	Mohammadi et al., 2017 (BM) [13]	GK	1–2 cm vs. <1 cm		x2.1 (RRN); x4.8 (SRN)
	Remick et al., 2020 (BM) [42]	LINAC non-CK	>2 cm		10%
Volume of BMs	Nakamura et al., 2001 (Mixed) [50]	GK	0.67–3 cc; 3.2–8.6 cc; and 8.7–95.1 cc		3%; 7%; and 9%
	Han et al., 2012 [41]	-	22.4 cc (median)		38.8% for large BM, low doses (13.8 Gy)
	Prabhu et al., 2017 (BM) [43]	-	5.9 cc		17.2%
	Mohammadi et al., 2017 (BM) [13]	GK	>0.1 cc		x2.1 (RRN); x4 (SRN)
	Loo et al., 2020 (BM) [44]	-	-		HR 1.09, 95% CI (1.01–1.18); *p* = 0.02

PBT: primary brain tumors; BM: brain metastasis; AVM: arteriovenous malformations; RRN: radiographic radionecrosis; SRN: symptomatic radionecrosis.

## Data Availability

Data requests will be reviewed by the senior author (RK).

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
