# Peer review of "Dosimetric Impact of Lesion Number, Size, and Volume on Mean Brain Dose with Stereotactic Radiosurgery for Multiple Brain Metastases"

_cancers, 2023, doi:10.3390/cancers15030780_

Round 1

Reviewer 1 Report

Thank you very much for the opportunity to review this exciting manuscript.

I have just a few short questions.

Why did the authors choose the investigated groups of sizes to be divided in 5mm steps? Was there a reason why the authors did not investigate the number of lesions also divided by the sizes, for which different single doses are chosen/grade 3 neurotoxicity was described? E.g. up to 2cm / 2-3cm / 3-4cm? I do not want to question the approach in general: In my opinion, the manuscript would benefit from at least a brief sentence, why the 5mm steps were chosen.

I did not find any info on a maximum dose allowed inside the GTV. Was there a homogenic dose prescription performed? 

Author Response

We greatly appreciate the thoughtful feedback, and we have carefully addressed each of the points raised in the following point-by-point responses. We believe that the comments and our corresponding revisions have significantly improved the quality of the manuscript. It is our hope that it is now acceptable for publication.

Comment #1

Why did the authors choose the investigated groups of sizes to be divided in 5mm steps? Answer

As described in the methods section of the manuscript, four size sets of GTVs were contoured and grouped by lesion size: Group 1 (<5mm), Group 2 (5-10mm), and Group 3 (>10-15mm), and Group 4 (>15 mm). This is consistent with other series which have demonstrated stepwise changes in rates of outcomes (i.e. tumor control or radiation necrosis) but smaller group sizes than the traditional RTOG 90-05 cutoffs. For example, Miller et al. demonstrated differences in the risk of radiation necrosis by lesion size using the same cut-offs as this study (Miller JA et al. IJROBP 2016). Ultimately, the 5mm increments provide better dosimetric discrimination and help highlight any differences base on delivery technology. As can be seen from the results of the manuscript, there are clear differences in the total number of lesions that can be treated using 5 mm increments vs. 1 cm or 2 cm increments.

Comment #2

Was there a reason why the authors did not investigate the number of lesions also divided by the sizes, for which different single doses are chosen/grade 3 neurotoxicity was described? E.g. up to 2cm / 2-3cm / 3-4cm?

Answer

Although size thresholds of < 2cm, 2.1 – 3 cm, and 3.1 – 4 cm have been used in prior single fraction SRS trials, there is clearly a limitation with single fraction SRS with larger lesions. Multiple series have demonstrated poor local control rates (<50%) for lesions > 2 cm treated with single fraction SRS (Shiau CY et al. IJROBP 1997, Hasegawa T et al. Neurosurg 2003, Vogelbaum MA et al. J Neurosurg 2006, Molenaar R et al. Br J Neurosurg 2009). Therefore, we restricted this analysis to the treatment of lesions that would be treated with a single course of SRS and to evaluate the comparion using small increments of lesion size.

Comment #3

I do not want to question the approach in general: In my opinion, the manuscript would benefit from at least a brief sentence, why the 5mm steps were chosen.

Answer

Related to comment #1. We agree with this comment. This is addressed in the revised version of the manuscript.

Comment #4

I did not find any info on a maximum dose allowed inside the GTV. Was there a homogenic dose prescription performed?

Answer

In this study, the only treatment planning requirement for GK and CK was a prescription isodose line of at least 50%. In other words, the maximum allowable dose was twice the prescription dose. For GK, an isodose line was picked for each target to generate at least 99.5% coverage to the target. For CK, the dose prescription isodose line was picked so that all of the targets in the plan were receiving the prescription dose to at least 99.6% of the volume.  There was no effort to generate a homogenous plan (not usually possible in GK), only to minimize the Gradient Index (GI) and increase dose drop off outside of target and reduce normal brain dose during planning.

Thank you so much for the valuable comments regarding our paper.

Reviewer 2 Report

This manuscript contains useful information for readers especially who are practicing stereotactic brain radiosurgery and/or stereotactic brain radiotherapy.  However, there is still room for improvement. 

Specific Comment:

1) Describe how the brain CT image set was selected and its rationale.

2) In method, you better describe how exactly each GTV was selected (e.g., mm interval in size?) and located within the brain in detail (e.g., how close can 2 GTVs be?).

3) Is each GTV a perfect sphere?

4) In figures, it is not easy to distinguish which one is which, you may consider using symbols in different shapes as well. 

Author Response

We greatly appreciate the thoughtful feedback, and we have carefully addressed each of the points raised in the following point-by-point responses. We believe that the comments and our corresponding revisions have significantly improved the quality of the manuscript. It is our hope that it is now acceptable for publication.

Comment #1

Describe how the brain CT image set was selected and its rationale.

Answer

The CT used for this study was based on a clinical case where we treated more than 75 lesions and became the impetus for this study. The reference brain CT image set was selected as it represents an average head CT data set. The physical dimensions of the head, volume of brain, visualization of anatomical structures and spatial resolution (i.e < 1mm voxels) of CT images were well suited for SRS planning. Additionally, the CT image data set was free of metal artifacts.   

Comment #2

In method, you better describe how exactly each GTV was selected (e.g., mm interval in size?) and located within the brain in detail (e.g., how close can 2 GTVs be?).

Answer

GTVs were contoured mimicking clinical scenarios with some targets far apart and some clustered together within 1 mm distance. Targets were then uniformly distributed throughout the whole brain.  We agree with this comment and we have added figures and a table describing the distribution of the distance to the closest neighbor for each target.

Comment #3

Is each GTV a perfect sphere?

Answer

No, the GTV was not always a perfect sphere.  Some were more ellipsoid than others and are based on actual clinical targets. Volumes varied from spherical to ellipsoidal in shape resembling clinical targets.

Comment #4

In figures, it is not easy to distinguish which one is which, you may consider using symbols in different shapes as well. 

Answer

We agree with this comment. This is addressed in the revised version of the manuscript.

Thank you so much for the valuable comments regarding our paper.